# FSF: Applying Machine Learning Techniques to Data Forwarding in Socially Selfish Opportunistic Networks

**DOI:** 10.3390/s19102374

**Published:** 2019-05-23

**Authors:** Camilo Souza, Edjair Mota, Diogo Soares, Pietro Manzoni, Juan-Carlos Cano, Carlos T. Calafate, Enrique Hernández-Orallo

**Affiliations:** 1Institute of Computing, Federal University of Amazonas, Manaus 69080-900, Brazil; edjair@icomp.ufam.edu.br (E.M.); diogo.soares@icomp.ufam.edu.br (D.S.); 2Departamento de Informática de Sistemas y Computadores. Universitat Politècnica de València, 46022 Valencia, Spain; pmanzoni@disca.upv.es (P.M.); jucano@disca.upv.es (J.-C.C.); calafate@disca.upv.es (C.T.C.); ehernandez@disca.upv.es (E.H.-O.)

**Keywords:** opportunistic networks, machine learning, friendship, selfishness, routing

## Abstract

Opportunistic networks are becoming a solution to provide communication support in areas with overloaded cellular networks, and in scenarios where a fixed infrastructure is not available, as in remote and developing regions. A critical issue, which still requires a satisfactory solution, is the design of an efficient data delivery solution trading off delivery efficiency, delay, and cost. To tackle this problem, most researchers have used either the network state or node mobility as a forwarding criterion. Solutions based on social behaviour have recently been considered as a promising alternative. Following the philosophy from this new category of protocols, in this work, we present our “FriendShip and Acquaintanceship Forwarding” (FSF) protocol, a routing protocol that makes its routing decisions considering the social ties between the nodes and both the selfishness and the device resources levels of the candidate node for message relaying. When a contact opportunity arises, FSF first classifies the social ties between the message destination and the candidate to relay. Then, by using logistic functions, FSF assesses the relay node selfishness to consider those cases in which the relay node is socially selfish. To consider those cases in which the relay node does not accept receipt of the message because its device has resource constraints at that moment, FSF looks at the resource levels of the relay node. By using the ONE simulator to carry out trace-driven simulation experiments, we find that, when accounting for selfishness on routing decisions, our FSF algorithm outperforms previously proposed schemes, by increasing the delivery ratio up to 20%, with the additional advantage of introducing a lower number of forwarding events. We also find that the chosen buffer management algorithm can become a critical element to improve network performance in scenarios with selfish nodes.

## 1. Introduction

In the last five years, activities related to Opportunistic networks (OppNets) have grown, attracting the attention of the scientific community [1]. This is due to the potential of these networks, which could become the solution to provide communication support in areas with overloaded cellular networks, and in scenarios where a fixed infrastructure is not available, like remote and developing regions. Nodes in OppNets exploit any connectivity that arises from node movements to distribute the contents, being largely based on the store-carry-forward mechanism derived from the delay and disruption-tolerant Networking (DTN) paradigm [2].

In OppNets, at every contact opportunity, the sender tries to send its messages to some relay nodes having a good probability of meeting their destinations at some other places [3]. However, since the path from a source to a destination is intermittently connected, the conventional routing algorithms are not applicable [4]. Consequently, new routing algorithms are required to overcome the long delays and frequent disconnection problems [5]. The traditional key challenge of a routing algorithm in an OppNet is to build a reliable path between a pair of nodes, enabling message exchanges between them. An efficient technique to achieve this aim is by selecting relay nodes with good chances to meet the destination in their next contacts. It is easy to realize that the better this selection is, the greater the chances of successfully delivering a message will be. Obviously increasing the delivery ratio means that the routing algorithm contributes to improving the network performance.

Another important and essential key challenge of a routing algorithm is related to the detection of selfish nodes in real network scenarios. A node may autonomously decide whether to accept or not a custody transfer of an incoming message from another node and, consequently, node cooperation is not fully guaranteed and must not be taken for granted [6]. If the routing algorithm decides to forward a message to a selfish relay, the latter can refuse to receive it if the message destination is an unknown or unwanted node. Then, how should we proceed if the selected relay node is not willing to carry the message? With respect to selfishness, according to [7], nodes can be classified as socially or individually selfish. Basically, a node classified as individually selfish does not collaborate with any other node, which means that it does not accept messages from other nodes in the network. On the other hand, a node classified as socially selfish is willing to relay messages to the nodes within the same community, but it refuses to relay messages to the nodes outside its community [7]. Thus, these nodes will only relay messages for nodes with whom they have some social tie (friends or acquaintances). Thus, taking into account these assumptions, it is easy to realize that a better routing algorithm must be proposed by including the detection of selfish nodes and also finding out the set of nodes that are willing to collaborate with it.

Having these premises in mind, in this work, we propose the “FriendShip and Acquaintanceship Forwarding” (FSF) algorithm, which makes its routing decisions taking into account the social ties between the nodes, and both the selfishness and the device resources’ levels of the candidate node for message relaying. Firstly, to choose good relay nodes for a message, FSF uses the social ties between the nodes as a suitable criterion. The intuition behind this reasoning is that the stronger the social tie is between the nodes, the bigger the probability of meeting each other in the future will be. FSF divides the social ties between the nodes into three groups: their friends, their acquaintances, and unknown nodes. To discover the friendship between the nodes, FSF uses the self-report friendship information (if available), or a mechanism based on machine learning techniques. In this mechanism, taking advantage of data derived from an experiment performed in the real world, we built a database containing the history of meetings of every pair of friends/not friends participating in the experiment. Then, to find out the existence of friendship between a pair of nodes in other scenarios, we apply a machine learning classifier to the created database. To discover the node’s acquaintances, FSF introduces a metric entitled Node’s Acquaintanceship Metric (NAM), which is based on the similarity between the history of contacts of each pair of nodes in the network. Next, to check the selfishness of the message relay candidate, FSF uses a mechanism that combines a collaborative watchdog system with a reputation system based on a logistic function. Finally, before it takes the forwarding decision, and to take into account those cases in which the relay node may not accept to receive the message because of resource constraints, FSF checks the resources levels of the candidate relay. When a contact opportunity arises, FSF will forward a message if, and only if, the social ties between the nodes are strong (friends or acquaintances), the message relay is not assessed as selfish, and the resources of the relay’ device are not at critical levels. On the contrary, the message will not be forwarded.

To validate our proposal, we develop a set of experiments that include simulating the message delivery process in four realistic well-known opportunistic network scenarios, i.e., Cambridge [8], Reality [9], National Chengchi University (NCCU) [10], and Sassy [11]. We then compare the obtained results in terms of delivery rate, average delay, and cost, against other protocols found in the literature such as BUBBLE-RAP [12], Friendship Routing [13], Probabilistic Routing Protocol using History of Encounters and Transitivity (PROPhET) [14], and Epidemic [15]. Simulation results show that, when taking the selfishness on routing issues into account, our FSF algorithm outperforms previously proposed schemes, by increasing the delivery ratio up to 20%, with the advantage that it introduces a lower number of forwarding events. We also find out that the buffer management algorithm can become an important key parameter that may impact the network performance in scenarios with the presence of selfish nodes.

In summary, the main contributions of this paper are:We present a novel system to classify the friendship strength between nodes using machine learning classifiers. We also present a new metric to depict the node’s acquaintanceships.We design the FSF algorithm by also taking into account the resources of the relay device on the routing decisions.We propose a watchdog and a reputation system to detect the nodes’ selfishness behaviour.We evaluate and compare the proposed algorithm with respect to other well-known routing algorithms, and show its performance improvements in realistic scenarios.

The rest of this paper is organized as follows: in Section 2, we discuss some related works, while, in Section 3, we present the architecture used. Section 4 details the proposed FSF protocol for OppNet applications. In Section 5, we describe the experimental setup, while, in Section 6, we present and discuss the experimental results. Finally, in Section 7, we conclude the paper and refer to future works.

## 2. Related Works

Several forwarding algorithms are available in the OppNets literature. Zhu et al. [16] classify forwarding algorithms according to their forwarding paradigm into three categories: (i) message-ferry-based, (ii) opportunity-based, and (iii) prediction-based. Recently, protocols based on social relationships have emerged as a fourth category. The routing protocol proposed in this work belongs to this novel category.

Message-ferry-based routing algorithms carry out data distribution in a network by combining the store-and-forward mechanism with the use of additional nodes, called ferries, which act as data mules [16]. Many routing algorithms have embraced this forwarding paradigm [17,18,19,20], including Burns et al. [21] who proposes a routing algorithm based on the observed meetings between peers, and the visits of peers to specific geographic locations. Message-ferry-based routing algorithms share a common problem: the overhead and extra cost to control the ferries.

In opportunistic-based algorithms, the nodes exchange data with each other when a contact occurs, whenever they are within communications range. The Epidemic routing algorithm, proposed in [22], is an example of an algorithm based on this forwarding paradigm. This routing algorithm assumes that, when a pair of nodes is in the same coverage area, they send all the messages stored in their buffers to each other, therefore increasing the message delivery probability.

Based on this opportunistic-based paradigm, some researchers have explored the network behaviour to propose forwarding algorithms. For instance, Shaghaghian and Coates [23] have used some simplifying assumptions about the network behaviour to propose two forwarding algorithms. The main goal is to reduce the expected latencies from any node in the network to a particular destination in some situations. The authors take into account that forwarding algorithms for opportunistic networks should result in low-average latency, and in an efficient usage of network resources. Based on simulation results, the authors confirm that their proposed algorithms are able to improve both the latency and the delivery rate.

Inspired by the Nash bargaining solution, Li et al. proposes in [24] the GameR forwarding algorithm for OppNets. This algorithm uses a utility function derived from the estimated resource utilization ratio and the history delivery predictability. According to the results obtained, GameR uses the network resources efficiently, improving the delivery ratio and decreasing the overhead on nodes under resource-constrained situations. Other proposals [25,26,27,28] also based on this paradigm are included.

Prediction-based algorithms are a refinement of the opportunistic-based approach. These algorithms select the next-hop node based on some metrics. These metrics make use of characteristics such as the probability of message delivery to the destination node [14,29,30] or the encounters’ history of the receiving node [27,31,32].

Finally, several researchers have considered the use of some social characteristics in the design of new DTN routing protocols. In fact, most DTN applications include a huge number of mobile devices carried by humans, and, according to Zhu et al. [16], the behaviour representing the utilization of devices is better described by social network models. Among the social characteristics worth considering, we may include altruism [33], friendship [13,34,35], selfishness [35,36], centrality, and popularity [12].

Hui et al. proposed in [12] the Bubble Rap Forwarding protocol. This protocol takes the routing decisions by using the concepts of community and centrality. Essentially, it selects high centrality nodes and community members related to the message destination. In [13], Bulut et al. proposed the first routing algorithm based on friendship. To model the friendship among nodes, the authors considered the frequency of contacts, their longevity, and their regularity. The authors claim that two nodes are friends if they regularly have long and frequent contacts. Based on these assumptions, a new metric called Social Pressure Metric (SPM) is proposed to represent the social pressure motivating friends to visit each other and share their experiences. This technique forwards messages between nodes only if they have a strong friendship.

Li et al. proposed a routing algorithm based on node selfishness [36]. The Social Selfishness Aware Routing (SSAR) is a protocol that introduced selfishness considerations in DTN scenarios. This routing algorithm tries to compensate for performance loss by allocating resources (buffers and bandwidth) based on packet priority. The authors of that paper consider that selfishness issues should be integrated into new routing algorithms since people are socially selfish, that is, they are willing to forward messages to a limited number of persons, and this willingness depends on the strength of the relationship among them. SSAR forwards messages between nodes only if they have a social tie. For example, if the social bonding between node A and node B is strong, node A will always accept messages for node B.

In our work, similarly to the work mentioned above, we model friendships by taking into account some historical information about the contact between nodes, like duration, frequency, and time between contacts. In addition, we also consider the number of meetings outside the university/work since we believe such information is a good evidence of friendship between two nodes. In the real world, it is reasonable to suppose that the more meetings outside work and/or university by a pair of nodes, the stronger the friendship between them will be. Our FSF routing algorithm also takes into account the selfishness of the message relay candidate in forwarding decisions. Unlike the works above, we consider that, in several situations, a node A may not accept those messages whose destination node is node B for the sake of saving resources, no matter how strong their mutual friendship is. Besides that, in this work, we have also considered two types of selfish behaviour: those nodes that are selfish only under specific conditions (e.g., resources constraints), and those nodes behaving selfishly all the time.

Overall, social-based protocols have been proposed to improve the performance of message forwarding. However, these features have not been explored yet sufficiently. To bridge this gap, we propose the FSF algorithm, whose characteristics are detailed below.

## 3. Architecture Overview

The FSF forwarding decisions are based on both the social tie and selfishness between the nodes. To get information about these two characteristics, every network node uses two different systems: the social tie classifier system, and the selfishness assessment system.
**The social tie classifier system**. The main goal of this system is to update information related to the social ties between each pair of nodes in the network. The social ties can be classified into *friends*, *acquaintances*, and *unknown*. A node stores information about its social ties in two different lists: *my social ties*, which stores the own node’s social ties, and *global social ties* which stores the social ties from the other network nodes.With the aim of updating *my social ties* list, every node has a database containing information about their meetings with other nodes in the network. The database belonging to the node $n_{i}$ is divided into n−1 rows, where the row $n_{k}$ stores data related to the history of meetings between nodes $n_{i}$ and $n_{k}$. Initially, this database is empty. Upon each contact opportunity, both the nodes in contact update the information related to the meetings between themselves in their own databases. Based on this information, they can classify the social tie with every other node participating in the network (as friends, acquaintances, or unknown). Such classification is performed periodically every *T* seconds.The *global social ties* list is updated at every contact opportunity taking place after the first *T* seconds. In a contact opportunity between the nodes $n_{i}$ and $n_{k}$, they share information about their own social ties (*my social ties* list) with each other, and then they include this information in the corresponding entry in their *global social ties* list. More details about these two lists will be provided in Section 4.1.2.**Selfishness assessment system**—The main goal of this system is to update the node’s reputation in terms of selfishness behaviour. In this work, a node can be classified as *individually selfish*, *socially selfish*, and *non-selfish*. Initially, the nodes have the same reputation and no information about the network. The node’s reputation is updated according to their behaviour when they were selected as a relay node in the past. In such a situation, a node can behave in three ways: (i) it always accepts receiving the message, (ii) never accept the message, (iii) it accepts receiving the message only if it is not experiencing resource constraints. Upon each decision, nodes update their own reputation. The selfishness assessment system is divided into two main systems: a detection scheme, which is implemented as a collaborative watchdog system based on [37], and a reputation system, which is based on a logistic function. More details about these components will be provided in Section 4.2.

### 3.1. Network Model

The network is modeled as a set of N nodes, divided into three groups: I nodes behave as *individually selfish*, S nodes behave as *socially selfish*, and C nodes behave as *non-selfish* (N = I + S + C). We consider that nodes are not malicious, that is, they do not send false information over the network (information about their own selfishness level). In the evaluations performed, we consider two scenarios: (i) evaluating the impact of increasing the message Time-To-Live (TTL), and (ii) evaluating the impact of increasing the buffer size. In the first scenario (i), we consider that the nodes have unlimited buffer size, that is, they can store all their messages and also the ones addressed to nodes with whom they want to collaborate. When the message TTL expires, it is automatically discarded from the node’s buffer. In the second scenario (ii), we consider that the messages have unlimited TTL with limited buffer, that is, the nodes can store a message for as long as they wish.

### 3.2. FSF Forwarding Strategy

When a contact opportunity arises, FSF will forward a message if, and only if, the nodes are friends or acquaintances (strong social tie), the message relay is not selfish or it is socially selfish, and its device has no resource constraints. On the contrary, the message will not be forwarded. For example, if a node A having a message M1 addressed to node B meets with node C, node A will forward M1 to node C if, and only if, the following assumptions are true: nodes B and C are friends or acquaintances of each other, node C is non-selfish or socially selfish, and the device resources of node C are not at critical levels. It is worth mentioning that, if node C is socially selfish, it will receive the message M1 only if B belongs to its friend’s circle. Otherwise, the message is not forwarded. In addition, if node A forwards the message to node C, the message will not be deleted from the buffer of node A as a way to increase the message delivery probability. Finally, if the nodes B and C are unknown to each other, or node C is assessed as individually selfish, or the device resources of node C are at critical levels, the message will not be forwarded. Figure 1 shows the flowchart representing the FSF forwarding strategy.

## 4. Detailed Design

In this section, we provide more details on how FSF classifies the social ties between the nodes, and how it assesses the relayer’s selfishness.

### 4.1. Social Tie Classifier System

The FSF system divides the node’s social ties into three groups: *friends*, *acquaintances*, and *unknown* nodes. The main difference between the nodes’ *friends* and *acquaintances* is that friends represent a stronger social tie than the node’s *acquaintances*. For example, in a university classroom, some nodes can be friends of a node $n_{x}$. These nodes are those with whom $n_{x}$ has a close relationship. Most of the time, they also meet each other outside the university. However, there are nodes with whom $n_{x}$ has no direct relationship, but they are frequently in contact. For example, to go to the university, $n_{x}$ usually takes the same bus, and it meets the same driver (and other people) every day. Despite they are not friends, they have regular contacts. It is also worth mentioning that messages addressed to nodes’ friends have the highest priority compared to those addressed to acquaintances, that is, FSF forwards those messages first. Furthermore, this division is important because, when a relay node is assessed as socially selfish, it will only accept receiving those messages addressed to its friends.

The social tie classifier system is divided into two mechanisms:**Nodes’ acquaintance mechanism**—it is a mechanism responsible for discovering the nodes’ acquaintances group. To achieve this aim, we propose a new metric to depict the node’s acquaintanceship.**Nodes’ friend mechanism**—it is the mechanism responsible for discover the nodes’ friends group. This mechanism takes into account two scenarios: (i) the information about the friendship strength between the nodes is available (self-reported friendship), and (ii) the self-reported information is not available.

As mentioned above, every network node stores information about the social ties in two lists entitled *my social ties* and *global social ties*.
*my social ties*—This list is divided into N−1 rows, one for each other node in the network. The rows have the format (id,our_social_tie), where our_social_tie can be *friends* (1), *acquaintances* (0), and *unknown* (−1). At every contact opportunity, both the nodes in contact update the row corresponding to each other in *my social ties* list.*global social ties*—This is a matrix containing *N* rows and *N* columns with 0’s in the diagonal, where the value stored at the position [*i*, *j*] represents the social tie between nodes *i* and *j* (1, 0, or −1). Every node uses this structure to store the already known information about the social ties between the other nodes in the network. At each contact opportunity, after updating the information, the nodes involved exchange their *my social ties* lists. Then, based on this information, they also update their *global social ties* matrix. Finally, based on this matrix, the nodes are able to build the first part of their own routing table.

As referred to above, the social ties’ discovery process is performed the first time after *T* seconds. The time between 0 and T−1 represents the period of time in which every node is moving around the network and collecting the data necessary for the first execution of the *Social tie classifier system*. After this initial period, every *T* seconds a new classification of the social ties is performed by every network node. Next, we detail the mechanisms used by the *Social tie classifier system*.

#### 4.1.1. Nodes’ Acquaintances Mechanism

To discover the node’s acquaintances, FSF introduces a metric called Node’s Acquaintanceship Metric (NAM). The NAM metric extends the metric similarity proposed by Li et al. [38]. The main goal of this metric is to take advantage of people’ daily routines to discover their acquaintances. It is largely known that people’s pattern mobility can be useful to determine when a pair of nodes will meet each other in the future [39,40]. NAM tries to take advantage of this by looking at the similarity of the contacts between a pair of nodes.

To the best understanding of the reader, consider the example shown in Figure 2. In this figure, we can see the history of encounters in four different weeks between a pair of nodes (id 13 and 22) participating in a real experiment entitled Sassy [11]. According to their self-reported friendship, these nodes are not friends, despite meeting each other frequently. It is worth mentioning that the history of contacts among them every week has similar values for some characteristics, like the duration, frequency (five days a week), and the time between contacts, among others. Based on this history of encounters, it is reasonable to suppose that there is a good chance that they will meet each other again along the following week. In the real world, it is easy to realize that this behaviour commonly occurs in people’ daily routines. For example, this behaviour can be applied to colleagues at work or at university, and to people sharing the bus or the metro every day, among others.

Based on these assumptions, the NAM metric takes into account that a pair of nodes $n_{x}$ and $n_{y}$ will be acquaintances of each other if, and only if, the history of contacts among them at times $T_{i}$ and $T_{i+1}$ are similar. To do this, we use the history of encounters among $n_{x}$ and $n_{y}$. To measure the similarity between the contacts of a pair of nodes, we have used the following characteristics: number of contacts (NC), contact duration (CD), time between contacts (TBC), and the most common time of contacts (MCTC).These specific characteristics can be estimated as follows:(1)NC(x,y)=n,
where *n* is the number of encounters among the nodes (*x*,*y*) at every period of time *T*,
(2)$$CD\;\left(xy\right)=\frac{\sum_{i=1}^{n}\left(t_{end(i)}-t_{start(i)}\right)}{n},$$
where $t_{start(i)}$ and $t_{end(i)}$ are the start time and the end time of the contact *i* between nodes *x* and *y*,
(3)$$TBC\;\left(xy\right)=\frac{\sum_{i=1}^{n}\left(t_{start(i+1)}-t_{end(i)}\right)}{n},$$
where $t_{start(i+1)}$ is the next contact start time, and $t_{end(i)}$ is the end time of the previous contact. To decrease the impact of data variability, for both the contact duration and the time between contacts, NAM takes the standard deviation into account. The last characteristic used by NAM is the most common time of encounters. To calculate it, NAM first stores the time when a specific contact takes place. Then, it uses the bimodal mode procedure to determine the most common hour for the contacts between $n_{x}$ and $n_{y}$. In order to select this type of mode, we have performed some experiments in the mobility traces used in our work. From the results, we have noted that the biggest number of modes in each collected sample is two. However, most of the time the samples used has just one mode. In the cases in which the samples have two modes, the FSF algorithm takes into account the lowest one. If there is no mode in the sample, our algorithm does not take into account this characteristic when calculating the metric NAM.

Lastly, to determine how similar the contacts are between $n_{x}$ and $n_{y}$, NAM uses a metric called cosine similarity. Let $u_{i}$ and $v_{i}$ be the vectors containing the history of contacts at times $T_{i}$ and $T_{i+1}$, respectively. NAM calculates the similarity between $u_{i}$ and $v_{i}$ as follows:(4)$$similarity_{u_{i},v_{i}}=cos\left(\theta \right)=\frac{u_{i}\times v_{i}}{\left|\right|u_{i}\left|\right|\times \left|\right|v_{i}\left|\right|},$$
where $\left|\right|u_{i}\left|\right|$ and $\left|\right|v_{i}\left|\right|$ represent the dot product of the vectors $u_{i}$ and $v_{i}$. To determine if a pair of nodes are acquaintances of each other, we set up a threshold γ of 0.7. If the value of Equation (Equation 4) for the nodes $n_{x}$ and $n_{y}$ is non-zero, and it is greater than or equal to γ, they are acquaintances of each other.

For example, taking into account the history of encounters shown in Figure 2, and data from the weeks 1 ($u_{i}$ = (5, 2, 22, 10)), and 2 ($v_{i}$ = (5, 1.5, 22, 10)), the FSF nodes’ acquaintanceship discovery will rely on the following steps:Calculate $u_{i}\times v_{i}$ = (5×5)+(2×1.5)+(22×22)+(10×10)=611,Calculate $\left|\right|u_{i}\left|\right|$ = ${\left(\left(5^{2}\right)+\left(2^{2}\right)+\left(22^{2}\right)+\left(10^{2}\right)\right)}^{0.5}=24.75,$Calculate $\left|\right|v_{i}\left|\right|$ = ${\left(\left(5^{2}\right)+\left(1.5^{2}\right)+\left(22^{2}\right)+\left(10^{2}\right)\right)}^{0.5}=24.71,$Calculate cos(θ) according to Equation (Equation 4) = $\frac{611}{\left||24.75||\times ||24.71|\right|}=0.99.$

Thus, in this example, nodes $\left|\right|u_{i}\left|\right|$ and $\left|\right|v_{i}\left|\right|$ are acquaintances of each other. Finally, in this work, the difference between $T_{i}$ and $T_{i+1}$ depends on the scenario. Table 1 includes the *T* values used in the different scenarios evaluated.

#### 4.1.2. Nodes’ Friends Discovering Mechanism

After discovering the nodes’ acquaintanceship, the social tie classifier system discovers the nodes’ friends. To obtain this information, our system takes into account two scenarios: (i) the information about the friendship strength between the nodes is available (self-reported friendship), and (ii) the self-reported information is not available.

In scenario (i), every node only loads this information in its *my social ties* list. This information can be found in some datasets derived from real experiments. For example, in the Reality Mining experiment [9], the authors provided to the scientific community diverse information from the users, including the self-reported friendship among them. Figure 3 shows the friendship graph used by FSF in the Sassy scenario [11].

In scenario (ii), the social tie classifier system uses a *machine learning scheme* to discover the existence of friendship between each pair of nodes in the network. This scheme is composed of a classifier algorithm and a database used for training the classifier. The classifier used is the Naive Bayes classifier [41]. We choose this classifier based on our previous works [35,42], where it achieved promising performance.

The Naive Bayes classifier uses the Bayes’ theorem to calculate the probabilities necessary to classify a new instance. From a machine learning perspective, we can define the Bayes’ theorem as follows: given an unknown instance $A=(a_{1},\ldots a_{n})$, where $a_{i}$ is one of the values of *A*, we wish to predict to which class *A* belongs.

According to Bayes’ theorem, the probability of choosing a class given a sample is given by:(5)$$P\left(Class|A\right)=\frac{P(Class|A)\times P(Class)}{P(A)}.$$

We can rewrite Equation (Equation 5) in terms of the attributes of instance A:(6)$$P\left(Class|A\right)=\frac{P\left(Class|a_{1},\ldots ,a_{n}\right)\times P\left(Class\right)}{P(a_{1},\ldots ,a_{n})}.$$

The class definition of instance A involves the computation of the probability for all possible classes given a particular attribute. The defined class is the one with the highest probability. From a statistical perspective, this is equivalent to maximizing P(Class |$a_{1}$, Class |$a_{2}$, ..., Class |$a_{n}$). The denominator in Equation (Equation 6) is constant, which leads to the following simplification:(7)$$P\left(Class|A\right)=argmax\;(P\left(Class|A\right)\times \prod_{i}P\left(Class|a_{1},\ldots a_{n}\right)).$$

To build the database used for training and testing the Naive Bayes classifier, we took advantage of data derived from the MIT research experiment [9]. In this experiment, the data were taken from a test with mobile devices equipped with a Bluetooth network interface. The experiment collected data from 97 mobile phones over the course of six months. We can have access to information related to connectivity, proximity, location, and activities of these users. Additionally, by answering the question “Is this person part of your close circle of friends?”, users assigned to which other people participating in the experiment they have a friendship relationship. Then, we built a database combining this self-reported friendship information with other attributes related to the users’ history of meetings.

In this way, this database is composed of the following attributes and their possible values:**Number of meetings** (NM)—Represents how many times two nodes were within the radio range of each other. Possible values: low, average, and high.**Average Contact duration** (ACD)—Represents the average time that two nodes remain within each other’s coverage area. Possible values: low, average, and high.**Average Time Between Contacts** (ATBC)—Represents the average time between the last contact and the new contact. Possible values: low, average, and high.**Meetings outside the university** (MOU)—Represents the information about the meetings between a pair of nodes outside the university. Possible values: yes (they meet outside the university), no (they do not meet outside the university).**Common Acquaintances** (CA)—Represents information about the common acquaintances between nodes. Possible values: yes (they have common acquaintances), no (they do not have common acquaintances).**Friendship strength** (FS)—Represents the self-reported friendship between a pair of nodes participating in the experiment. From a machine learning perspective, it is the class. Possible values: strong (they are friends), and weak (they are not friends).

In this database, the values for the attributes NM, ACD, and ATBC correspond to the history of meetings between each pair of nodes from the MIT experiment at the above-mentioned time *T*. In order to categorize the values for these attributes in the levels described above (low, average, high), we performed two steps. Firslty, we have used algorithms from machine learning to preprocessing the database used as training for the classifier. Then, we have used three percentile groups (25th, 75th, 100th). Then, let NM values from the MIT experiment be represented by numerical values (0 to 50 encounter times). We create three levels of NM, and convert these numerical values into NM levels. For example, from 0 to 4 encounters, the frequency is tagged as low, from 5 to 15, it is considered average, and above 15 it is considered high. In turn, we model the attribute CA by using the aforementioned node’s acquaintances discovery mechanism. We choose the attributes NM, ACD, and ATBC based on other works found in the literature [13,35]. We also included the attributes CA and MOU because we believe that they are representative parameters for classifying the friendship between a pair of nodes. Finally, once again, the *T* value depends on the scenario (refer to Table 1). Table 2 shows an example of some tuples used in this work as training data by the Naive Bayes algorithm.

#### 4.1.3. Example of Naive Bayes Classification

To clarify our approach, we present an example. Let the database used as the training set for the Naive Bayes classifier resemble the data presented in Table 2, and consider that when the above-mentioned time *T* takes place, the collected history of meetings between a pair of nodes *(X,Y)* is represented by the instance *A = (NM = high, ACD = high, ATBC = low, MOU = yes, CA = no, FS?)*. To answer if *(X, Y)* are friends or not, FSF asks the Naive Bayes Algorithm, which will follow the steps below:

**Step 1**—Determine the probability P(Class) of occurrence of each class (Equation (Equation 5)). P(Class) results from the division of the number of cases of class *X* by the total number of tuples in the database represented in the Table 2:


$P\left(Class=strong\right)=\frac{5}{15}=0.34,$



$P\left(Class=weak\right)=\frac{10}{15}=0.66,$


**Step 2**—Determine the probability $P(Class|a_{1},\ldots ,a_{n})$ of the attributes for the instance in question about every possible class (Equation (Equation 6)):


$P\left(Class=strong|NM=high\right)=\frac{3}{5}=0.6,$



$P\left(Class=weak|NM=high\right)=\frac{2}{10}=0.2,$



$P\left(Class=strong|ACD=high\right)=\frac{2}{5}=0.4,$



$P\left(Class=weak|ACD=high\right)=\frac{4}{10}=0.4,$



$P\left(Class=strong|ATBC=low\right)=\frac{1}{5}=0.2,$



$P\left(Class=weak|ATBC=low\right)=\frac{5}{10}=0.5,$



$P\left(Class=strong|MOU=no\right)=\frac{4}{5}=0.8,$



$P\left(Class=weak|MOU=no\right)=\frac{5}{10}=0.5,$



$P\left(Class=strong|CA=no\right)=\frac{2}{5}=0.4,$



$P\left(Class=weak|CA=no\right)=\frac{5}{10}=0.5,$


**Step 3**—Determine the probability of each class based on the probability of the instance (Equation (Equation 7)):


P(Class=strong|A)=(0.6×0.4×0.2×0.8×0.4×0.34)=0.0005,



P(Class=weak|A)=(0.2×0.4×0.5×0.5×0.5×0.66)=0.0066.


In this example, and according to the calculated probabilities, the friendship strength between nodes *(X,Y)* would be classified as *weak*. It is worth mentioning that, in this system, a pair of nodes can be classified either as an acquaintance or as a friend. Taking into account that this classification can have an impact on the FSF forwarding decisions, the system considers in such case that the pair of nodes are friends because it is the most important social tie. Finally, if a pair of nodes are not classified either as friends or acquaintances, they are considered unknown nodes.

### 4.2. Selfishness Assessment System

Before taking the forwarding decision, FSF assesses the selfishness of the message relay candidate. To detect selfishness, this system takes into account the nodes’ behaviour when they were selected as a relay node in the past. In such situations, a node can behave in three different ways:**individually selfish**—according to the selfishness classification made in [7], individual selfishness can be defined as “the unwillingness of a single node to relay the messages of all other nodes to conserve its limited resources”. These nodes do not accept to relay messages.**socially selfish**—it can be defined as a type of selfishness in which a node belongs to a community, and it is willing to relay messages to the nodes within the same community, but it refuses to relay messages to the nodes outside its community [7]. Thus, these nodes will only relay messages for nodes with whom they have some social tie (friends or acquaintances).**non-selfish**—those nodes accepting to relay messages to any other node.

The selfishness assessment system is composed of two mechanisms: the detection scheme and the reputation system. Figure 4 shows the interaction between these components in the selfishness system. The detection scheme is implemented as a watchdog system. According to [43], watchdog systems are components installed in every network node, and that are responsible for overhearing data communication between nodes to decide about the neighbours’ selfishness. In this work, we have used a collaborative watchdog system based on [37]. When a neighbour node has exhibited a selfish behaviour in the past, it tends to return a positive detection. On the contrary, it assesses the node as being non-selfish. However, despite a watchdog system has been proved as an interesting technique to detect the nodes’ selfish behaviour, its decisions can be impacted by the problem known as outliers’ detection. According to [44], outliers are normally defined as data points that have a significant difference from the rest of the data according to a certain measure. In this way, to decrease the impact of outliers, we have used a reputation system based on [45]. In this reputation system, upon each contact, one assessment is performed, even when the nodes are not selfish. Node A can get information from a neighbour B in the following two situations:
**(i)** **Selfish Contact:**Node A, using its monitoring mechanism during a contact, detects that a neighbour node B is selfish.**(ii)** **Non-Selfish Contact:**Both nodes A and B are non-selfish. Thus, they may share with each other information about the selfishness of their neighbours.

It is worth mentioning that, if the detection scheme collects an update about a neighbour, the reputation system is updated. The node reputation is defined taking into account the probability of being more cooperative than other nodes. This probability is based on logistic functions, as shown in Equation (Equation 8) [45]:(8)$$P_{cooperation}\left(A,B\right)=\frac{1}{1+10^{\frac{R_{B}-R_{A}}{F_{d}}}},$$
where $P_{cooperation}\left(A,B\right)$ describes the probability of node *A* to be more cooperative than node *B*. $R_{A}$ and $R_{B}$ represent the reputations of both nodes *A* and *B*, respectively, and $F_{d}$ is a significant factor to stress the difference between the reputations of the nodes (A,B). The node reputation update process is defined as a two-step process for each pair of contacts (A,B): (i) check node *B* behaviour, and (ii) update node *B* reputation. The process of updating the reputation works as follows:
**(i)** **Average Reputation**At every contact, the node calculates its average reputation using Equation (Equation 9), that is, the arithmetic mean of the reputation of the neighbours at the moment of contact with node *B*:
(9)$$AvgR\left(A\right)=\frac{\sum_{\forall k\in neighborhood_{A}-\left\{B\right\}}R_{k}}{|neighborhood_{A}|-1}.$$**(ii)** **Node Update**When a selfish node is detected, a node updates its reputation by using Equation (Equation 10), where D(B)∈{0,1} assumes 0 when *B* is detected as selfish, and 1 otherwise. δ is the weight assigned to each new observation:
(10)$$R_{B}{}^{\prime }=R_{B}+\delta \times \left(D\left(B\right)\right)-P_{cooperation}\left(A,B\right)).$$

By using this reputation model, each node can infer the behaviour of a neighbour node based on its reputation value.

Additionally, we also consider that a node (socially selfish or non-selfish) can refuse to receive a message due to resource depletion. The intuition behind this reasoning is that, in the real world, when the devices have resources constrained, the message relay could not accept to receive messages to save its resources. To model this situation, FSF considers two devices’ resources levels represented by two different thresholds: the energy level (represented by α), and the memory space (represented by β). When a contact opportunity arises, if the message relaying device has an energy level less than α, or if the memory space used is more than β, the node will decline to relay the message, independently of whether it is socially selfish or non-selfish. To decide which specific values should be assigned to these thresholds, we have performed a performance evaluation of FSF by variating the values used by these thresholds. We compared the FSF performance in terms of the delivery ratio and the delivery delay, in the scenarios evaluated in this work. We variated α from 10% to 80%, and β from 20% to 90%. Based on the results achieved, we find that, for α greater than 30% and β greater than 70%, the results are very similar. Thus, in our simulations, these thresholds were set by using these values.

## 5. Experimental Setup

In this section, we provide more details about the experimental setup used to evaluate our proposal.

### 5.1. Extensions to the ONE Simulator

To evaluate our FSF algorithm, we performed trace-driven simulations using the ONE Simulator [46]. We implemented within the ONE Simulator our FSF algorithm. We included two new classes in the simulator, entitled *FriendshipClassifier* and *SelfishnessAssessment*. These two classes implement the mechanisms shown in Section 4.1 and Section 4.2, respectively. To use the machine learning mechanism described in Section 4.1.2, we integrated the simulator with the WEKA tool [47]. This is a powerful tool implemented in Java, which contains the implementations of several machine learning algorithms. We also made modifications to the already existent classes *MessageRouter*, *ActiveRouter*, *DTNHost*, and *DTNSim*. On the other hand, the algorithms used for comparison, the event generators (mobility readers and message generation), and the reports used for analysis, are those available in The ONE Simulator package.

### 5.2. Nodes’ Mobility

To simulate node mobility, we used the following well-known real mobility traces:**Cambridge** [8]—In the Cambridge trace, the devices were distributed to students from the University of Cambridge, mainly to students attending the Computer Laboratory, specifically Ph.D. and Master students and undergraduates.**Reality** [9]—In Reality, mobile phones were distributed to 97 students from the MIT and used over the course of six months.**NCCU** [10]—By using an Android application installed on smart handheld devices, it represents 115 students’ moving on a typical school day at the National Chengchi University over a period of 15 days.**Sassy** [11]—In this experiment, the authors distributed 27 devices to students at the University of St. Andrews over a period of 79 days.

Table 1 lists some parameters describing the traces used in our simulation experiments.

### 5.3. Routing Algorithms Used for Comparison

For the sake of comparison, we selected the following routing algorithms:**Epidemic [15]**—For this algorithm, every contact is an opportunity for the nodes to exchange all the messages stored in their buffers with each other. It is worth mentioning that, despite it achieving good performance, its routing strategy consumes many network resources. However, it is an interesting benchmark for comparison.**PROPhET [14]**—This algorithm is based on determining the probability of meeting with the message destination. A node carrying a message will send it to another node if the delivery predictability of the destination of the message is higher at the other node. For every contact oportunity, this algorithm compares the probability of each node meeting the message destination node in the future. The node carrying a message will forward it to the other node if the latter has a higher meeting probability.**BUBBLE-RAP [12]**—This algorithm takes into account the concepts of nodes’ popularity and centrality in the routing decisions. It selects high centrality nodes and community members related to the message destination node.**Friendship routing [13]**—This is the first algorithm that takes into account the friendship strength between the nodes in the routing decisions. It forwards messages to relay nodes having a greater friendship with the message destination.

It is worth mentioning that these algorithms were not designed taking into account the existence of selfishness nodes in the network. In this way, in our simulations, we modified these algorithms to also consider the selfishness between the nodes. To achieve this aim, each of these algorithms takes advantage of our selfishness assessment system described in Section 4.2. Thus, if the relay node selected by them is assessed as a selfish node at that moment, the message is refused. Otherwise, they use their original routing decisions.

### 5.4. Metrics of Interest

Based on previous works in the literature [12,13,14], we have used the following metrics in our performance evaluations:**Delivery ratio**—it can be defined as the ratio of messages received by the destination nodes to those generated by the source nodes.**Average delivery delay**—it can be defined as the average time interval between the sending and receiving events for messages travelling along the network.**Average cost**—it can be defined as the average number of forwarding events per message delivered to the destination.

### 5.5. Nodes’ Power Consumption

To simulate the power consumption of nodes, we adopted the energy model proposed in [48]. The power consumption of a node is classified into five states:**Off**—no power consumption as the network node interface is turned off.**Inactive**—reduced power consumption as the network node interface is idle.**Scan**—the node consumes power while the network node interface detects neighbours.**Transmission**—the node consumes power while sending a message.**Reception**—the node consumes power because it is receiving a message.

In our simulations, we assume that all the nodes initially have the greatest level of energy. We consider that the energy level of nodes is measured in units, being 500 units the highest value. We assume that users recharge their device every 24 h. Energy consumption depends on the state of the device and the number of operations using the network interface. For example: if the node is at transmission, reception or scan states, we assume a reduction of 25 energy units for every sent/received message and/or for every scan performed. If the node is in the off or inactive states, we assume there is no energy cost.

## 6. Results and Discussion

We carried out trace-driven simulations to compare the FSF performance against the algorithms cited on the Section 5.3 by considering two different scenarios: (i) Increasing values of the message TTL, where the nodes have unlimited buffer size, that is, they can store their own messages, as well the messages addressed to other nodes with whom they accept to collaborate. When the TTL expires, FSF automatically drops the associated message from the buffer; (ii) Increasing buffer size, where the nodes have unlimited TTL, that is, there is no time limit for storing a message, but buffers have limited space storage.

Following similar works found in the literature, message size varies from 512 KB to 1 MB, while buffer size varies from 10 MB to 60 MB, and TTL values varies from 5 to 96 h. The simulator used a uniform distribution to generate 1000 messages between a pair of nodes. Each experiment determined a 95% confidence interval for the estimated parameters. Both scenarios adopted the Drop Oldest algorithm [42] as the buffer management strategy.

### 6.1. Results Depending on the Message TTL

Figure 5, Figure 6 and Figure 7 present the results when increasing values of message TTL. By using social relationships among the nodes as a forwarding criterion, we intuitively expect to improve data exchange by natural encounters. The results shown in Figure 5 confirm this intuition, since the FSF algorithm achieved the best delivery ratio in all scenarios, delivering up to 7% more messages than the other algorithms. It is worth mentioning that, in all tested scenarios, the algorithms behave quite similarly, that is, the larger the message TTL is, the higher the delivered message ratio is. In addition, the Friendship Routing algorithm gets the second-best delivery ratio in all scenarios tested, confirming that the use of social ties (friendship, acquaintances, bridges, etc.) is an interesting criterion to increase the message delivery probability in OppNets.

With respect to the metrics delivery delay and average cost, the algorithms exhibit the same behaviour: the higher the message TLL, the longer the delivery delay, the higher the average cost. In addition, for all algorithms, the higher the delivery ratio, the higher the delivery delay is. Since the FSF algorithm delivered more messages, it also gets the the longest delivery delay. Regarding the metric average cost, FSF achieved the best result in Cambridge and NCCU. The Bubble-Rap gets the best result in Sassy, while, in Reality, the PRoPHET algorithm performed better.

Almost certainly, the number of relationships among the nodes, as well as the intercontact time between nodes with the social relationship, has impacted the FSF performance. The higher the average number of social relationships per node, the higher is the probability of forwarding a message during a contact opportunity. It clearly can also increase the FSF average cost. On the other hand, when the intercontact time between nodes with the social relationship is high, it clearly can impact the average delivery delay. From the simulation results, it is reasonable to assume that the average number of social relationships detected by FSF for each node is low and the intercontact time is high, which contributes to increasing the FSF average delay and reducing the number of messages forwarding.

### 6.2. Results Depending on the Buffer Size

Figure 8, Figure 9 and Figure 10 present the results when increasing values of the buffer size. Regarding the delivery ratio, in all tested scenarios, the algorithms behave quite similarly, that is, the larger the buffer size, the higher the delivery ratio. The FSF algorithm outperformed the other algorithms in the Cambridge and Sassy scenarios, achieving ratios of up to 18% and 5%, respectively. The Friendship Routing algorithm, also based on social characteristics, gets the second-best delivery ratio in all scenarios. The PRoPHET routing strategy achieved the best delivery ratio in the NCCU scenario (up to 20%), followed by Epidemic routing.

Again, the algorithms which delivered more messages took more average time to deliver these messages. For example, FSF in the Cambridge and Sassy scenarios, PRoPHET in the NCCU, and Friendship Routing in the Reality scenario. With respect to the metric average cost, FSF also outperformed the other algorithms in all scenarios tested. Moreover, it is worthwhile to emphasize that FSF also achieved the highest delivery ratio in the Cambridge, Sassy, and Reality scenarios. Altogether, the above results highlight the benefits of the FSF strategy for improving the delivery ratio in opportunistic networks, besides reducing the resource utilization.

### 6.3. Understanding the FSF Performance

To better understand the factors impacting the performance of our FSF algorithm, we implemented and analysed three variants of the FSF algorithm:**FSF with social-based Buffer Management (FSF+BM)**—We want to evaluate the impact of using a buffer management algorithm to favour the FSF forwarding. According to [49], the message scheduling and discarding strategy can strongly impact the routing performance in OppNets. When the nodes buffers are limited, nodes can soon operate close to their maximum storage capacity. When node A tries to forward a message, but there is no space in the receiving buffer of node B, B should drop one or more stored messages. The best choice is to preserve the messages with a high probability of being forwarded in future contact opportunities. Moreover, during a contact opportunity, every node must decide which messages it should forward. This is a crucial issue since the contact duration cannot be long enough to the exchange of all messages they would like to. Thus, we selected the social-based algorithm called Friendly-Drop [50] as the buffer management algorithm.**FSF without Energy Constraint (FSFwEC)**—We want to evaluate the impact of power constrained nodes over the FSF performance. In forwarding events, nodes consume more energy, and thus their energy level is not enough to take part in the network. In this scenario, we selected the Drop Oldest as the buffer management algorithm.**FSF without Selfish Nodes (FSFwSN)**—We want to evaluate the impact over the FSF performance when there are no selfish nodes. The nodes may refuse to accept messages if they are under resource constraints. In this scenario, we selected the Drop Oldest as the buffer management algorithm.

The rest of the simulation parameters are the same as presented in Table 1.

Figure 11, Figure 12 and Figure 13 depict the simulation results when we increased the message TTL in the four tested mobility scenarios. The algorithms exhibit an increasing delivery ratio when compared to the original FSF implementation. The lack of selfish nodes led to a higher delivery ratio. With no power constraints, more messages arrive at their destination, making it clear that, when operating near to their energy capacity, nodes increase their selfish behaviour, and hence having an impact on the delivery ratio. Although the FSF+BM did not achieve the best results, one can note a higher delivery ratio compared to the original FSF implementation.

Regarding the delivery delay and the average cost, compared to the original implementation, the behaviour of the algorithms was similar: the higher the message TTL is, the longer the average delay and the higher the average cost (e.g., the Cambridge and NCCU scenarios). With respect to the delivery delay, a reasonable justification for this result relates to the increase in the delivery ratio achieved by the algorithms tested. On the other hand, the absence of selfish nodes and energy constraints contributes to increasing the average forwarding number required to successfully deliver a message. In the original FSF implementation, a reasonable number of nodes is selected as a relay, but those nodes may refuse to accept the message because they are selfish or they are facing energy constraints at that moment. It clearly decreases the amount of forwarding, as well as the probability of message delivery.

Figure 14, Figure 15 and Figure 16 summarize the simulation results when increasing the buffer size. Once again, the delivery ratio of the FSFwSN was higher, while the FSFwEC achieved the second best result. Notice that, again, all the implementations achieved a higher delivery rate when compared to the original FSF implementation. It is worth highlighting that, in scenarios like Reality and Sassy, there is no significant difference in the performance of FSF+BM and FSFwEC.

Regarding the delivery delay and the average cost, the behaviour did not change: the larger the buffer size, the longer the average message delivery delay and the average cost. Again, a reasonable justification relates to the increase in the delivery ratio. The algorithm FSFwSN had a lower average delivery delay, and a lower average cost as well.

The results show that the presence of selfish nodes strongly impacts the performance of the FSF algorithm and the other ones found in the literature. However, taking into account the results of FS + BM, it is reasonable to assume that the management of factors like buffer and energy can mitigate the impact of selfish nodes.

### 6.4. Additional Discussions

One important issue that affects the performance of the FSF algorithm is the correct classification of the social ties between nodes in the network. With respect to the friendship, in this work, we have introduced an approach to detect this relationship by using a machine learning aproach. This algorithm has two important tasks: first, learn about the friendship between nodes based on data collected from the real world. Second, classify new relationships between two nodes of the network. Based on the results of previous work, we selected the well-known Naive Bayes, as it can achieve a promising performance to rank the friendship between nodes. The better the classification performed by the Naive Bayes algorithm, the better the performance of the FSF algorithm.

The implementation of a machine learning algorithm proved to be an interesting solution to the task of detecting the existence of friendship between two nodes. However, some issues should be considered. For example, the availability of information about what friendship is in the real world may be an important requirement because, in some scenarios, this information is not available. To solve this problem, one can monitor the scenario through an application that would be responsible for collecting information about the friendship between the nodes. Another alternative is to use a machine learning algorithm that does not need a training database to classify new instances.

Finally, we consider that the use of a machine learning approach may be more flexible than other approaches to classifying the friendship between the nodes. For example, if there is a need to change the characteristics used in the friendship model, using machine learning can accelerate this process. Another advantage is that a machine learning technique can combine several characteristics to rank the friendship, and thus can easily use data from real-world situations.

## 7. Conclusions and Future Work

In this work, we presented an approach that includes the use of social relationships as a criterion for disseminating messages in Opportunistic Networks. We propose the “FriendShip and Acquaintanceship Forwarding” (FSF) algorithm that considers two social characteristics: the social relationships between nodes, and the selfishness of relay candidate nodes. FSF performs two main tasks: first, FSF classifies the social bonds between nodes; second, FSF uses a reputation system to verify the egoism of the relay node by considering cases where, despite a strong social relation to the destination, the relay node may refuse to receive the message because it is selfish, or its device has resources’ constraints at that time.

To validate the proposed algorithm, we performed a set of experiments to determine the effectiveness of message delivery in four scenarios based on trace-based simulations by means of the ONE simulator. Compared to other routing algorithms of the same class, FSF achieved better results in the context of a set of standard metrics, namely the delivery rate, the average cost, in addition to achieving reasonable results in terms of delivery delays. In addition, we also conducted a set of experiments to determine the impact of three FSF variants. From the results obtained, it can be observed that FSF performance is strongly impacted by the presence of selfish nodes in the network. However, the results also showed that adding a buffer management algorithm can improve the delivery ratio, even in scenarios with the presence of selfish nodes in the network.

As future work, we intend to carry out a comparative study of adopting other machine learning algorithms to classify the friendship strength among pairs of nodes. Moreover, we want to analyze the impact of selfishness in the routing process by using different thresholds for device resource restrictions.

## Figures and Tables

**Figure 1 sensors-19-02374-f001:**
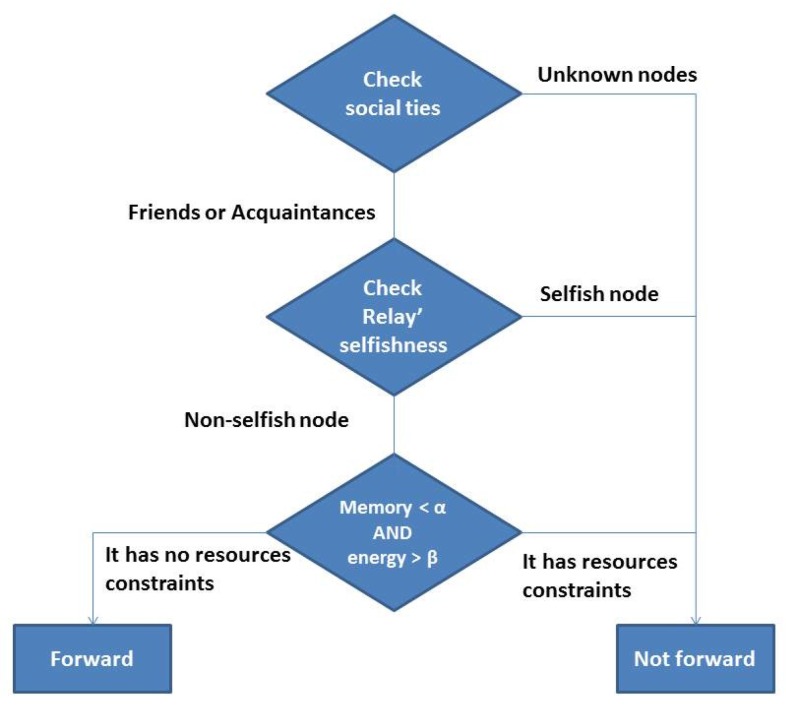
Flowchart of the FSF forwarding strategy.

**Figure 2 sensors-19-02374-f002:**
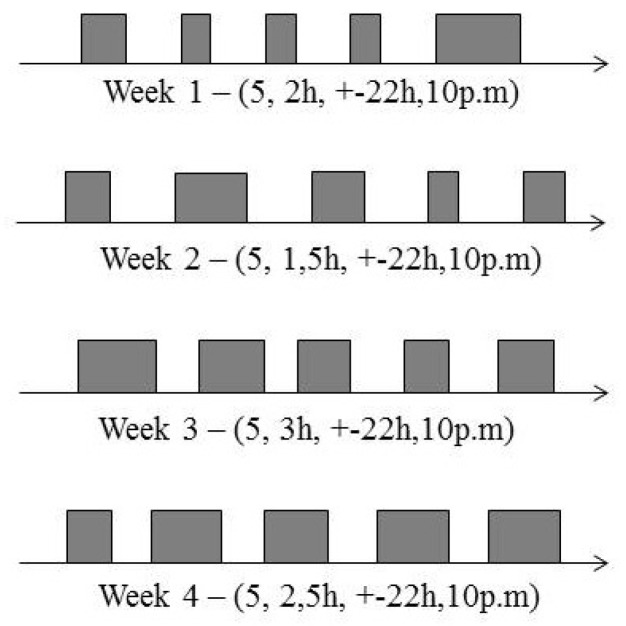
History of encounters between nodes 13 and 22 in the Sassy Scenario. According to the self-reported friendship, they are not friends. However, they meet each other frequently, as shown above.

**Figure 3 sensors-19-02374-f003:**
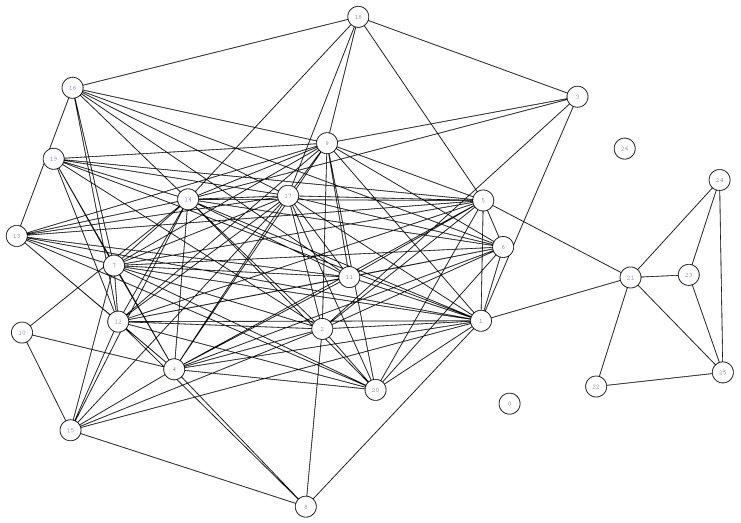
Self-reported friendship graph from the nodes participating in the Sassy experiment.

**Figure 4 sensors-19-02374-f004:**
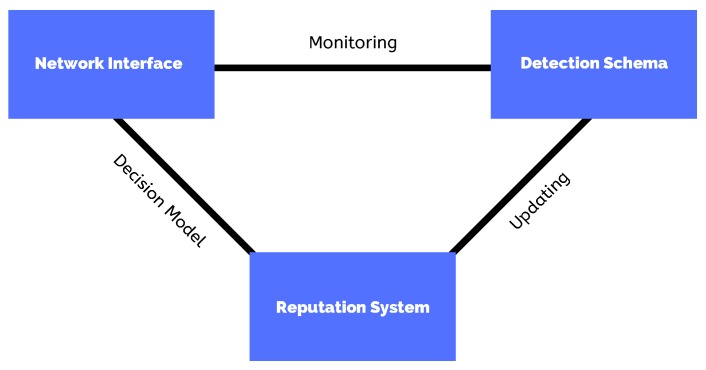
The selfishness assessment system running at each network node.

**Figure 5 sensors-19-02374-f005:**
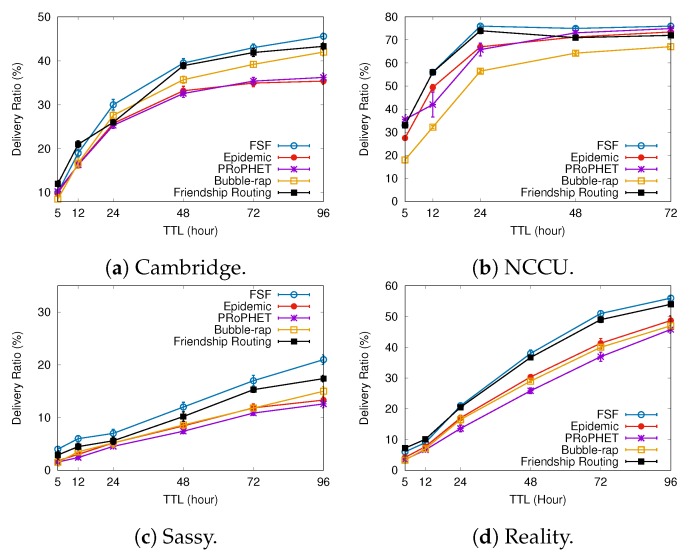
Delivery ratio when increasing message TTL.

**Figure 6 sensors-19-02374-f006:**
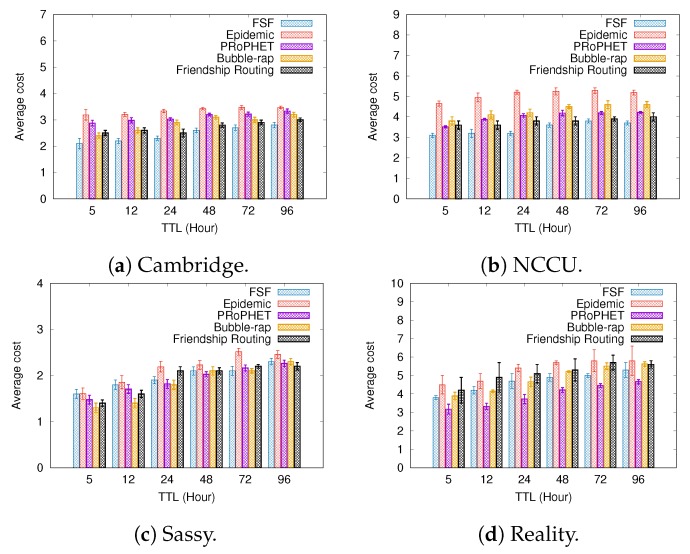
Average cost when increasing message TTL.

**Figure 7 sensors-19-02374-f007:**
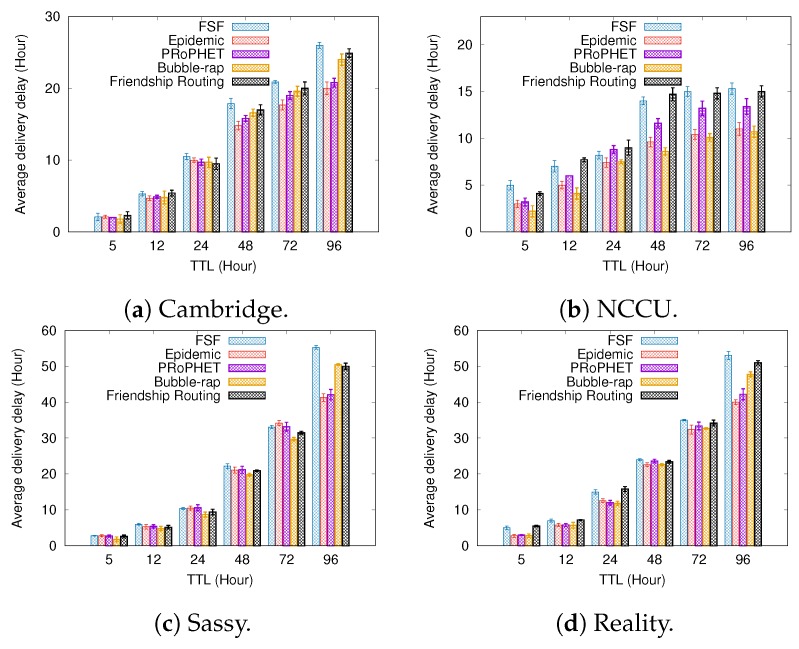
Delivery delay when increasing message TTL.

**Figure 8 sensors-19-02374-f008:**
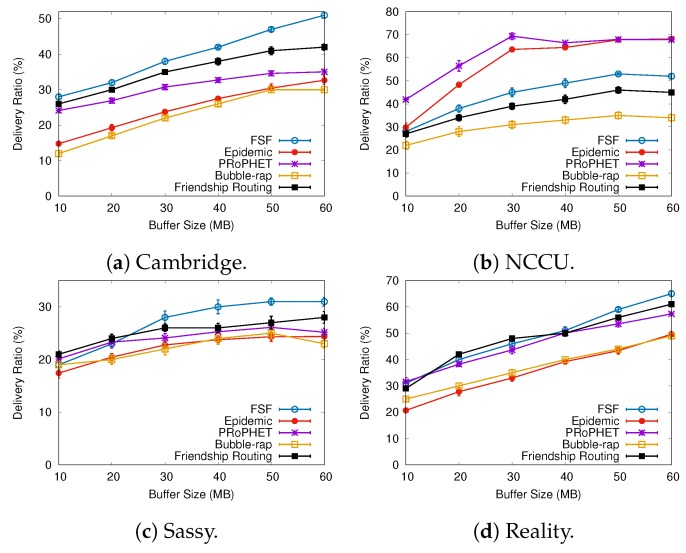
Delivery ratio when varying the buffer size.

**Figure 9 sensors-19-02374-f009:**
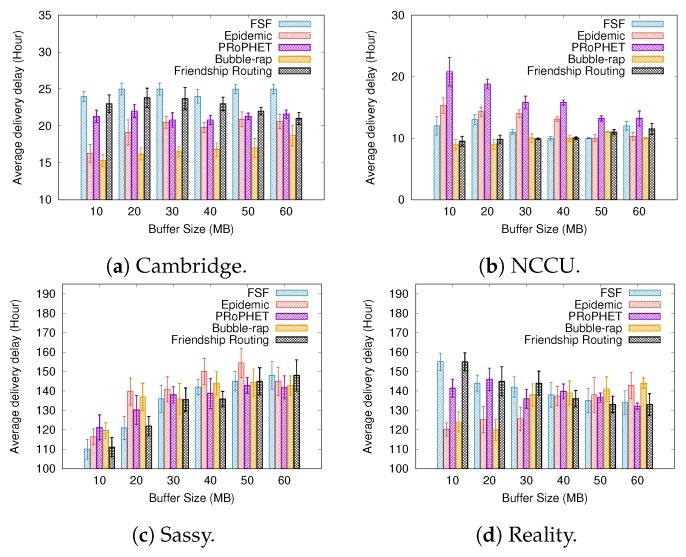
Delivery delay when varying the buffer size.

**Figure 10 sensors-19-02374-f010:**
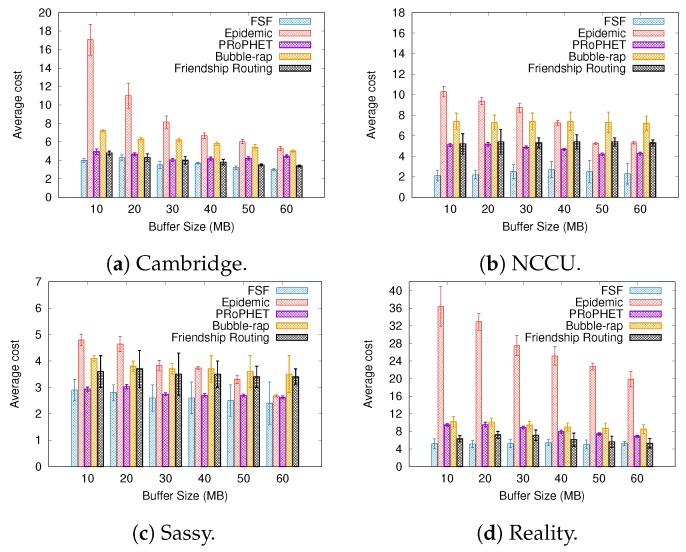
Average cost when varying the buffer size.

**Figure 11 sensors-19-02374-f011:**
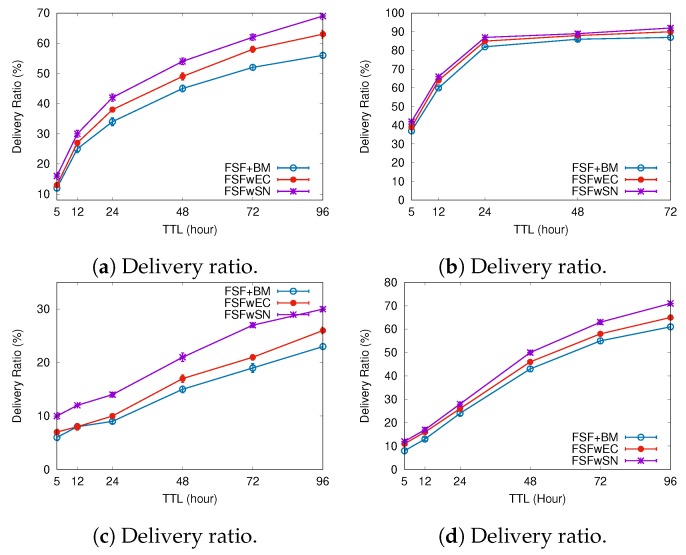
Delivery ratio when increasing the message TTL.

**Figure 12 sensors-19-02374-f012:**
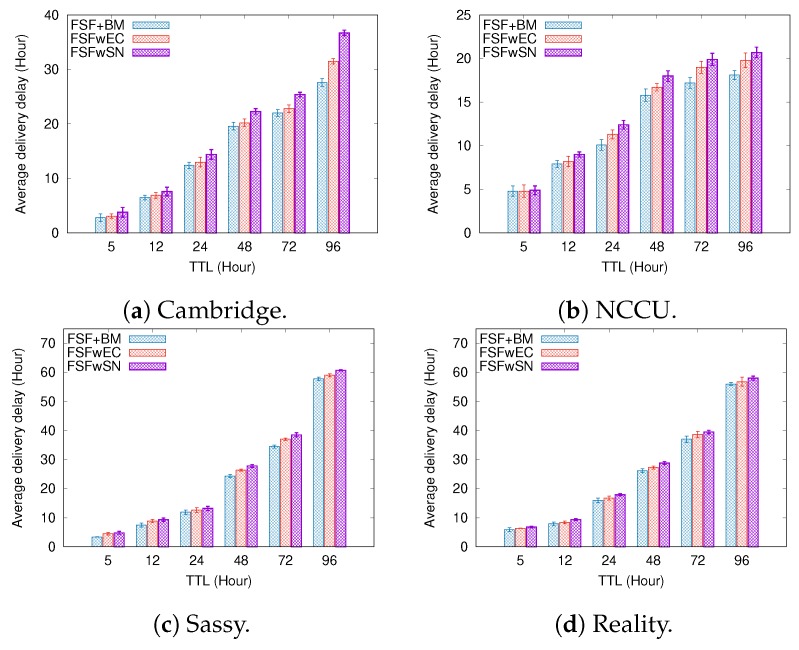
Delivery delay when increasing the message TTL.

**Figure 13 sensors-19-02374-f013:**
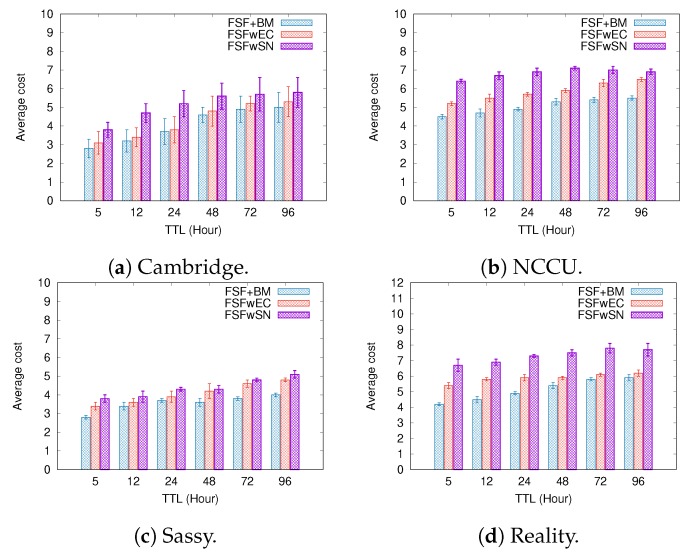
Average cost when increasing the message TTL.

**Figure 14 sensors-19-02374-f014:**
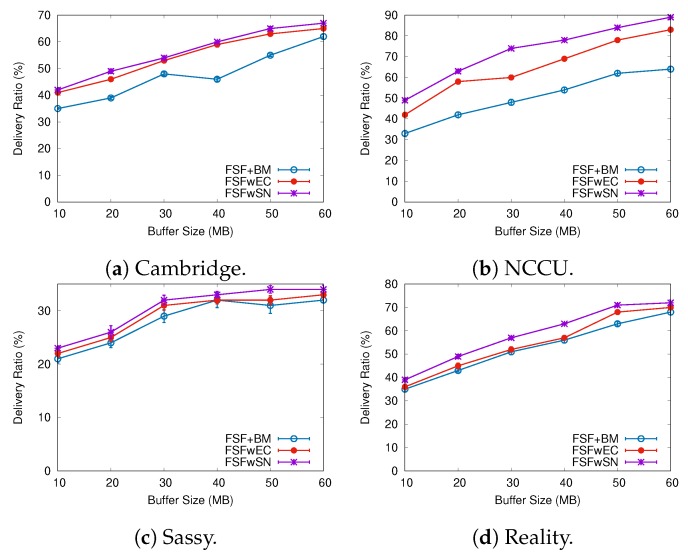
Delivery ratio when varying the buffer size.

**Figure 15 sensors-19-02374-f015:**
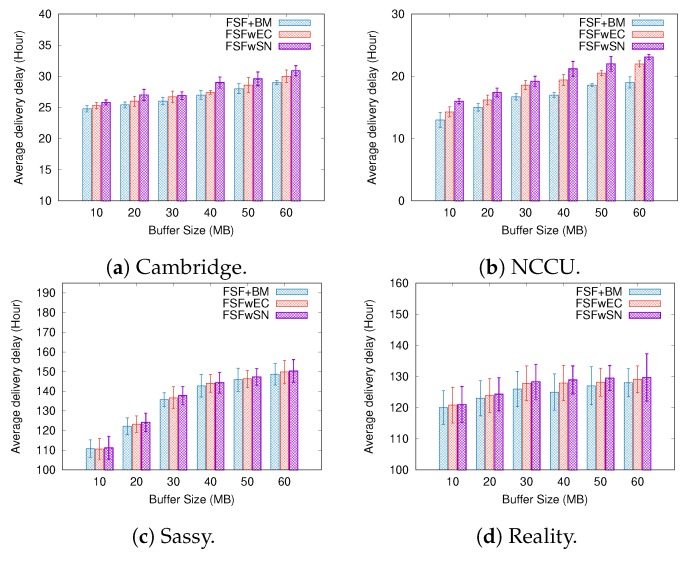
Delivery delay when varying the buffer size.

**Figure 16 sensors-19-02374-f016:**
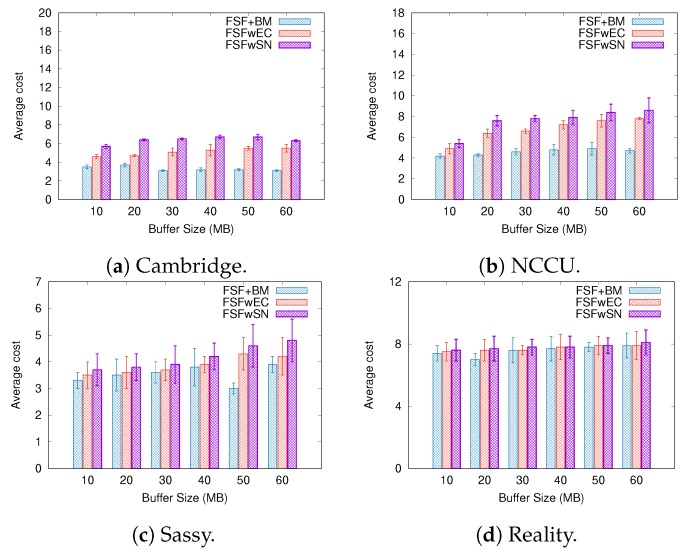
Average Cost when varying the buffer size.

**Table 1 sensors-19-02374-t001:** Parameters describing the traces used in the simulation.

Trace	*Cambridge*	*Reality*	*NCCU*	*Sassy*
**Device**	iMotes	Phone	Smart Handheld	T-mote
**Network Interface**	Bluetooth	Bluetooth	Wi-fi/Bluetooth	Sensor
**Nodes**	54	97	115	27
**Trace Duration** (days)	11	246	15	79
**Number of contacts** (aprox.)	10.873	54.667	81.115	35.274
**T value** (days)	3	7	3	7

**Table 2 sensors-19-02374-t002:** A sample of the training data used in this work.

NM	ACD	ATBC	MOU	CA	FS
low	high	average	yes	no	weak
low	high	high	no	yes	weak
average	average	average	no	yes	weak
high	average	low	yes	yes	weak
high	low	low	no	no	strong
high	low	average	no	yes	strong
average	high	low	yes	no	weak
low	average	low	yes	no	weak
high	average	average	no	yes	strong
average	low	high	no	no	weak
average	high	high	no	yes	strong
high	high	high	yes	yes	weak
low	high	average	yes	no	strong
average	average	low	no	no	weak
low	low	low	no	yes	weak

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
