# Peer review of "FSF: Applying Machine Learning Techniques to Data Forwarding in Socially Selfish Opportunistic Networks"

_sensors, 2019, doi:10.3390/s19102374_

Reviewer 1 Report

In your paper, you presented a routing protocol "FriendShip and Acquaintanceship Forwarding" (FSF) considering the social ties between the nodes and their possible selfishness. Having read your paper, I would like you to comment on the following:
- On page 2, lines 34/35, you write that the key challenge of a routing algorithm in an OppNet is to "build a reliable path" between a disconnected pair of nodes. But if these are disconnected, such a path cannot exist.
- You assume that a node can detect the selfishness of another node. Is this always possible? What if the node simply accepts all messages for forwarding, but the silently deletes them?
- What is the effort for the user? As far as I have understood, the user needs to define which other nodes belong to a friend. This might be very difficult as the nodes have cryptic IDs which are hard to map to a person.
- My idea of the epidemic protocol is to
- Your approach collects and processes a huge amount of information, which you totally neglect in your simulation results. Please specify the amount of data each meeting pair of nodes has to exchange before they can forward any message. And also estimate the complexity of the learning algorithm and of the processing of all the tables your approach requires. This definitely will lead to a high power consumption that you did not consider in section 5.5.
- Please check the references to the figures in the text. In my opinion, there is a problem starting with the reference to figure 5 on page 12.
- My knowledge of Epidemic is that the nodes forward a message to all the other nodes they meet and keep a copy. This means that the message should always arrive after the shortest possible delay. In your simulations, it looks like Epidemic is worse than most of the other approaches. Could you please comment on this?
- Having scanned through your references, I found that only 14 from 51 are younger than 5 years. Does this mean that the research topic is not a recent one?

Author Response

Dear revisor, thanks for your valuable review. We answer your questions below:
- Comment 1 - On page 2, lines 34/35, you write that the key challenge of a routing algorithm in an OppNet is to "build a reliable path" between a disconnected pair of nodes. But if these are disconnected, such a path cannot exist.

R: In fact, there is a mistake in that phrase. We have changed this sentence. Please refer to the new version of our paper to check it.

- Comment 2 -

2.1 - You assume that a node can detect the selfishness of another node. Is this always possible?

R – In order to detect selfishness behavior, our algorithm uses data related to the behavior of the users in the past with respect to their decisions about receiving some message or not. The intuition behind this is that, if a user has declined to receive a message in the past, there is a high probability that he/she did that because he/she is selfish. Then, taking into account this information our algorithm can assess the reason for that behavior. Our algorithm will be unable to detect the selfishness behavior in two situations: i) if the user is not interested in participating in the network, which means that he/she will not send/receive messages in the network, and ii) if the user does not give the permissions necessary to collect and use your personal data. Otherwise, our algorithm can take the information necessary to perform its evaluations. 

2.2 - What if the node simply accepts all messages for forwarding, but the silently deletes them?

R – The user can do this and obviously, it will impact the performance, mainly the delivery ratio, but this behavior is beyond the scope of our current selfishness system. The main goal of our selfishness system is to detect the probability of the user to be selfish or not, and using this information do the routing decisions. In other words, our system is just prepared to help to take the decisions before the forwarding event. All happening after that is not being used to feed our system. We can interpret that if the user does that, he/she can be assessed as a selfish node, but additionally, he/she is acting in bad faith too, that is known as a malicious behavior. We believe it could be an interesting extension of our system to cope with these malicious nodes. Thanks for the observation.

Comment 3 - What is the effort for the user? As far as I have understood, the user needs to define which other nodes belong to a friend. This might be very difficult as the nodes have cryptic IDs which are hard to map to a person.

R – Our social ties system was prepared to work if the information about the social relationships about the nodes is available or not. When this information is available, no evaluation is executed, our system just uses these relationships as a basis to take the routing decisions. On the other hand, if this information is not available, our system collects data to evaluate and classify the social ties between the nodes. We believe that the node identification can be done by using some information like the MAC address, IMEI, Serial number, or by combining all this information.
Comment 4 -  Your approach collects and processes a huge amount of information, which you totally neglect in your simulation results. Please specify the amount of data each meeting pair of nodes has to exchange before they can forward any message. And also estimate the complexity of the learning algorithm and of the processing of all the tables your approach requires. This definitely will lead to a high power consumption that you did not consider in section 5.5.

R – For the sake of clarity, we now include this information into the paper.

Our analysis can be divided into two parts:

4.1 – Social ties system: in order to take its decisions, this system uses two structures: a list and a matrix. The size of them is directly proportional to the number of nodes that one interacts with. In this analysis, take into account that:

- Every position of the list has 2 attributes: (an iD and a social tie) both are represented by integers.

- Every position of the matrix has just one attribute: a social tie, also represented by an integer.

- In Java, an integer contains 4 bytes.

Now, consider that a node interacts with 100 other nodes. The list will have 100 elements and the global matrix will have 100 lines x 100 columns. In this way, the size of the list is:

Size of the list: 100 elements x 4 bytes x 4 bytes = 160 bytes or 0.16 Kb

Size of the Matrix: 100 x 100 x 4 bytes = 40,000 bytes, or 40 Kb.

Considering a 2Mbps Bluetooth link, for example, it only needs 0.16 seconds to send the matrix. We believe this is a reasonable time.

With respect to this point, we also need to take into account two facts:

- From the results presented in the paper, we can see that FSF decreases the number of forwardings. In this way, it can contribute to decreasing the impact caused by sending the matrix used by FSF.

- We believe that most of the time each node does not interact with more than 1000 other nodes. Then, it means that the matrix   will be in all cases lower than 40 Kb.

Thus, we believe these facts minimize the impact caused by the needed of sending the matrix.

With respect to the complexity, in both cases, since we have the information about the iDs, the complexity to access the information is:

List: O(N) in the worst case, because our algorithm needs to find the element in the list.

Matrix: O(1) because the indices are available and our algorithm does not need to search any item, just access it directly and change its value.
Comment 5 - Please check the references to the figures in the text. In my opinion, there is a problem starting with the reference to figure 5 on page 12.

R: You are right. We have fixed now this issue. Thanks for your detailed review.

Comment 6 - My knowledge of Epidemic is that the nodes forward a message to all the other nodes they meet and keep a copy. This means that the message should always arrive after the shortest possible delay. In your simulations, it looks like Epidemic is worse than most of the other approaches. Could you please comment on this?

R: In fact, this is the Epidemic strategy. However, in order to understand the Epidemic performance, we need to take into account the following points:

- In the evaluated scenarios we consider selfish nodes. It means that even if a node running the Epidemic routing decides in sending a message to another node, the candidate to relay can refuse to receive it if he/she is a selfish node.

-  In the evaluated scenarios we also consider that a node can refuse to receive a message if his/her device is running close to their energy capacity. Since that Epidemic strategy increases the resources consumption, it is reasonable to suppose that several messages are dropped because of this.
Comment 7 - Having scanned through your references, I found that only 14 from 51 are younger than 5 years. Does this mean that the research topic is not a recent one?

R – Yes, it is.  Firstly, as you comment, 14 of the cited papers have less than 5 years, so in absolute terms, the research topic can be considered a hot topic. With regard to the other older references, we have cited these papers because they fit with our research and in our opinion they help to make the paper self contained.

p { margin-bottom: 6.25px; direction: ltr; color: rgb(0, 0, 0); line-height: 115%; }p.western { font-family: "Liberation Serif", "Times New Roman", serif; font-size: 16px; }p.cjk { font-family: "AR PL SungtiL GB"; font-size: 16px; }p.ctl { font-family: "Lohit Devanagari", "Times New Roman"; font-size: 16px; }

Reviewer 2 Report

The authors propose an interesting work.

However, there are still some issues worthy improved before considering its publication.

1)Technical issues: 

a.Although the authors show the performance differences between their method and the others, they did not precisely explain the reason why the relevant thresholds, parameters, or both used in their FSF should or would be so done. For example, as noted by the authors in lines 464-466 of page 13, “To decide which specific value should be assigned to these thresholds, we performed an evaluation varying the thresholds \alpha and \beta. …”, they perform an evaluation to decide the thresholds, but do not show the evaluating mechanism and its correctness. Further, while the authors can perform an evaluation for the thresholds, the other parameters, such as the levels of (low, average, high) represented by the percentile groups (25th, 75th, 100th), could be evaluated similarly to show their optimality for this work. Otherwise, it would be interesting to know if the levels represented by, e.g., (25th, 50th, 100th), would be better than the above. Besides, at page 8, the authors claim that it uses the statistical mode procedure to determine the most common hour for the contacts between n_x and n_y. It is also interesting to know the statistical mode procedure they exactly use. Apart from these examples, the other parameters are all worth to note or explain more rigorously with the reasons why they should be given the values currently shown.

b.In this work, every node maintains two list entitled my social ties and global social ties. Does the mechanism save the communication overhead when compared with the others? As noted in page 7, at each contact, after updating, the nodes will exchange my social ties lists, and then, they also update their global social ties matrix. If so, the overall information would be eventually updated at each contact, which could bring a significant communication overhead. 

2)Writing issues:

a.Some similar/duplicated sentences or descriptions can be found throughout the manuscript. For example, the sentence in lines 248-249 of page 7 is similar to that in lines 273-274 of the same page. 

b.Some abbreviations are given inconsistently. For example, Friendship Routing is first abbreviated as FR in page 2, but then abbreviated as FS in page 14.

c.Some notations are not given consistent or even not defined. For example, P_{coop}(A, B) in equation (8) seems to be P_{cooperation}(A, B) in equation (10), but they are represented differently. In addition, “RF” is not defined in equation (10), and \delta noted as the weight assigned to each new observation is not actually shown in equation (10).

d.Some figure numbers are incorrect. For example, in line 327 of page 9, it should be Figure 3 rather than Figure 4. As another example, in line 427 of page 12, it should be Figure 4 instead of Figure 5.

As a summary, although the readers like me may be aware of these issues and would try to correctly interpret the context, these technical issues and writing problems would still damp the value of this work, and should be eliminated as possible.

Author Response

Dear revisor, thank you very much for your valuable review and comments. We answer your points below.

Comment 1) Although the authors show the performance differences between their method and the others, they did not precisely explain the reason why the relevant thresholds, parameters, or both used in their FSF should or would be so done. For example, as noted by the authors in lines 464-466 of page 13, “To decide which specific value should be assigned to these thresholds, we performed an evaluation varying the thresholds \alpha and \beta. …”, they perform an evaluation to decide the thresholds, but do not show the evaluating mechanism and its correctness. Further, while the authors can perform an evaluation for the thresholds, the other parameters, such as the levels of (low, average, high) represented by the percentile groups (25th, 75th, 100th), could be evaluated similarly to show their optimality for this work. Otherwise, it would be interesting to know if the levels represented by, e.g., (25th, 50th, 100th), would be better than the above. Besides, at page 8, the authors claim that it uses the statistical mode procedure to determine the most common hour for the contacts between n_x and n_y. It is also interesting to know the statistical mode procedure they exactly use. Apart from these examples, the other parameters are all worth to note or explain more rigorously with the reasons why they should be given the values currently shown.

R: We include in the new version of the paper information that makes this observation more clear. In fact, in order to categorize the values for the parameters Number of meetings, Average contact duration, and Average time between contacts, we have used the following methodology:

1 –  Our first step was to apply algorithms from machine learning to preprocessing the database used as training for the classifier.

2 –  After this, we have used the percentile function from Excel to select the levels for each parameter.

With respect to the thresholds alfa and beta, we have performed a performance evaluation of FSF by variating the values used by these thresholds. We compared the FSF performance in terms of the delivery ratio and the delivery delay, in the scenarios Reality and Cambridge of the following variations of FSF:

FSF with alpha = 80% and beta = 20%

FSF with alpha = 70% and beta = 30%

FSF with alpha = 60% and beta = 40%

FSF with alpha = 50% and beta = 50%

FSF with alpha = 40% and beta = 60%

FSF with alpha = 30% and beta = 70%

FSF with alpha = 20% and beta = 80%

FSF with alpha = 10% and beta = 90%

As cited in the paper, from alpha = 30% and beta = 70%, the results were very similar. Then, in our simulations, these thresholds were set by using these values.

With respect to the mode, we have used the bimodal mode. In order to select this type of mode, we have performed some experiments in the mobility traces used in our work. And in all of them, the biggest number of modes in each sample was 2. However, most of the time the samples used had just one mode. In the cases where the samples had 2 modes, our algorithm takes into account the lowest one. If there is no mode in the sample, our algorithm does not take into account this characteristic when calculating the metric NAM.

Comment 2) In this work, every node maintains two list entitled my social ties and global social ties. Does the mechanism save the communication overhead when compared with the others? As noted in page 7, at each contact, after updating, the nodes will exchange my social ties lists, and then, they also update their global social ties matrix. If so, the overall information would be eventually updated at each contact, which could bring a significant communication overhead. 

R – The nodes in contact exchange with each other the global social ties matrix, even if they have no messages to exchange with each other. When we were designing the FSF algorithm, we have performed an analysis to decide if the matrix would be sent in each contact or not.

Our analysis was divided into two parts:

4.1 – Social ties system: in order to take its decisions, this system uses two structures: a list and a matrix. The size of them is directly proportional to the number of nodes that one interact. In this analysis, take into account that:

- Every position of the list has 2 attributes: (an iD, a social tie) both are represented by integers.

- Every position of the matrix has just one attribute: a social tie, also represented by an integer.

- In Java, an integer contains 4 bytes.

Now, consider that a node interacts with 100 other nodes. The list will have 100 elements and the global matrix will have 100 lines x 100 columns. In this way, the size of the list is:

Size of the list: 100 elements x 4 bytes x 4 bytes = 160 bytes or 0.16 Kb

Size of the Matrix: 100 x 100 x 4 bytes = 40,000 bytes, or 40 Kb.

Considering a 2Mbps Bluetooth link, for example, it needs 0.16 seconds to complete the sending of the matrix. We believe this is a reasonable time.

With respect to this point, we also need to take into account two facts:

- From the results presented in the paper, we can see that FSF decreases the number of forwardings. In this way, it can contribute to decreasing the impact caused by sending the matrix used by FSF.

- We believe that most of the time each node does not interact with less than 100 other nodes. Then, it means that the matrix size is lower than 40 Kb.

Thus, we believe these facts minimize the impact caused by the needed of sending the matrix.

Writing issues:

All the writing issues were fixed. Please, refer to the new version of the paper to check it.

td p { margin-bottom: 0; direction: ltr; color: rgb(0, 0, 0); }td p.western { font-family: "Liberation Serif", "Times New Roman", serif; font-size: 16px; }td p.cjk { font-family: "AR PL SungtiL GB"; font-size: 16px; }td p.ctl { font-family: "Lohit Devanagari", "Times New Roman"; font-size: 16px; }p { margin-bottom: 6.25px; direction: ltr; color: rgb(0, 0, 0); line-height: 115%; }p.western { font-family: "Liberation Serif", "Times New Roman", serif; font-size: 16px; }p.cjk { font-family: "AR PL SungtiL GB"; font-size: 16px; }p.ctl { font-family: "Lohit Devanagari", "Times New Roman"; font-size: 16px; }

Round  2

Reviewer 2 Report

The authors have addressed the issues given previously.